# Optimization over Continuous and Multi-dimensional Decisions with Observational Data

**Dimitris Bertsimas**
Sloan School of Management
Massachusetts Institute of Technology
Cambridge, MA 02142
dbertsim@mit.edu

**Christopher McCord**
Operations Research Center
Massachusetts Institute of Technology
Cambridge, MA 02142
mccord@mit.edu

## Abstract

We consider the optimization of an uncertain objective over continuous and multi-dimensional decision spaces in problems in which we are only provided with observational data. We propose a novel algorithmic framework that is tractable, asymptotically consistent, and superior to comparable methods on example problems. Our approach leverages predictive machine learning methods and incorporates information on the uncertainty of the predicted outcomes for the purpose of prescribing decisions. We demonstrate the efficacy of our method on examples involving both synthetic and real data sets.

## 1 Introduction

We study the general problem in which a decision maker seeks to optimize a known objective function that depends on an uncertain quantity. The uncertain quantity has an unknown distribution, which may be affected by the action chosen by the decision maker. Many important problems across a variety of fields fit into this framework. In healthcare, for example, a doctor aims to prescribe drugs in specific dosages to regulate a patient's vital signs. In revenue management, a store owner must decide how to price various products in order to maximize profit. In online retail, companies decide which products to display for a user to maximize sales. The general problem we study is characterized by the following components:

- Decision variable: $z \in \mathcal{Z} \subset \mathbb{R}^p$,
- Outcome: $Y(z) \in \mathcal{Y}$ (We adopt the potential outcomes framework [20], in which $Y(z)$ denotes the (random) quantity that would have been observed had decision $z$ been chosen.),
- Auxiliary covariates (also called side-information or context): $x \in \mathcal{X} \subset \mathbb{R}^d$,
- Cost function: $c(z; y) : \mathcal{Z} \times \mathcal{Y} \to \mathbb{R}$. (This function is known *a priori*.)

We allow the auxiliary covariates, decision variable, and outcome to take values on multi-dimensional, continuous sets. A decision-maker seeks to determine the action that minimizes the conditional expected cost:

$$\min_{z \in \mathcal{Z}} \mathbb{E}[c(z; Y(z))|X = x]. \tag{1}$$

Of course, the distribution of $Y(z)$ is unknown, so it is not possible to solve this problem exactly. However, we assume that we have access to *observational data*, consisting of $n$ independent and identically distributed observations, $(X_i, Z_i, Y_i)$ for $i = 1, \ldots, n$. Each of these observations consists of an auxiliary covariate vector, a decision, and an observed outcome. This type of data presents two challenges that differentiate our problem from a predictive machine learning problem. First, it is incomplete. We only observe $Y_i := Y_i(Z_i)$, the outcome associated with the applied decision. We do

not observe what the outcome would have been under a different decision. Second, the decisions were not necessarily chosen independently of the outcomes, as they would have been in a randomized experiment, and we do not know how the decisions were assigned. Following common practice in the causal inference literature, we make the ignorability assumption of Hirano and Imbens [13].

**Assumption 1** (Ignorability).

$$Y(z) \perp\!\!\!\perp Z \mid X \quad \forall z \in \mathcal{Z}$$

In other words, we assume that historically the decision $Z$ has been chosen as a function of the auxiliary covariates $X$. There were no unmeasured confounding variables that affected both the choice of decision and the outcome.

Under this assumption, we are able to rewrite the objective of (1) as

$$\mathbb{E}[c(z; Y) \mid X = x, Z = z].$$

This form of the objective is easier to learn because it depends only on the observed outcome, not on the counterfactual outcomes. A direct approach to solve this problem is to use a regression method to predict the cost as a function of $x$ and $z$ and then choose $z$ to minimize this predicted cost. If the selected regression method is uniformly consistent in $z$, then the action chosen by this method will be asymptotically optimal under certain conditions. (We will formalize this later.) However, this requires choosing a regression method that ensures the optimization problem is tractable. For this work, we restrict our attention to linear and tree-based methods, such as CART [7] and random forests [6], as they are both effective and tractable for many practical problems.

A key issue with the direct approach is that it tries to learn too much. It tries to learn the expected outcome under every possible decision, and the level of uncertainty associated with the predicted expected cost can vary between different decisions. This method can lead us to select a decision which has a small point estimate of the cost, but a large uncertainty interval.

## 1.1 Notation

Throughout the paper, we use capital letters to refer to random quantities and lower case letters to refer to deterministic quantities. Thus, we use $Z$ to refer to the decision randomly assigned by the (unknown) historical policy and $z$ to refer to a specific action. For a given, auxiliary covariate vector, $x$, and a proposed decision, $z$, the conditional expectation $\mathbb{E}[c(z; Y)|X = z, Z = z]$ means the expectation of the cost function $c(z; Y)$ under the conditional measure in which $X$ is fixed as $x$ and $Z$ is fixed as $z$. We ignore details of measurability throughout and assume this conditional expectation is well defined. Throughout, all norms are $\ell_2$ norms unless otherwise specified. We use $(X, Z)$ to denote vector concatenation.

## 1.2 Related Work

Recent years have seen tremendous interest in the area of data-driven optimization. Much of this work combines ideas from the statistics and machine learning literature with techniques from mathematical optimization. Bertsimas and Kallus [4] developed a framework that uses nonparametric machine learning methods to solve data-driven optimization problems in the presence of auxiliary covariates. They take advantage of the fact that for many machine learning algorithms, the predictions are given by a linear combination of the training samples' target variables. Kao et al. [17] and Elmachtoub and Grigas [11] developed algorithms that make predictions tailored for use in specific optimization problems. However, they all deal with the setting in which the decision does not affect the outcome. This is insufficient for many applications, such as pricing, in which the demand for a product is clearly affected by the price. Bertsimas and Kallus [5] later studied the limitations of predictive approaches to pricing problems. In particular, they demonstrated that confounding in the data between the decision and outcome can lead to large optimality gaps if ignored. They proposed a kernel-based method for data-driven optimization in this setting, but it does not scale well with the dimension of the decision space. Misic [19] developed an efficient mixed integer optimization formulation for problems in which the predicted cost is given by a tree ensemble model. This approach scales fairly well with the dimension of the decision space but does not consider the need for uncertainty penalization.

Another relevant area of research is causal inference (see Rosenbaum [20] for an overview), which concerns the study of causal effects from observational data. Much of the work in this area has

focused on determining whether a treatment has a significant effect on the population as a whole. However, a growing body of work has focused on learning optimal, personalized treatments from observational data. Athey and Wager [1] proposed an algorithm that achieves optimal (up to a constant factor) regret bounds in learning a treatment policy when there are two potential treatments. Kallus [14] proposed an algorithm to efficiently learn a treatment policy when there is a finite set of potential treatments. Bertsimas et al. [3] developed a tree-based algorithm that learns to personalize treatment assignments from observational data. It is based on the optimal trees machine learning method [2] and has performed well in experiments. Considerably less attention has been paid to problems with a continuous decision space. Hirano and Imbens [13] introduced the problem of inference with a continuous treatment, and Flores [12] studied the problem of learning an optimal policy in this setting. Recently, Kallus and Zhou [16] developed an approach to policy learning with a continuous decision variable that generalizes the idea of inverse propensity score weighting. Our approach differs in that we focus on regression-based methods, which we believe scale better with the dimension of the decision space and avoid the need for density estimation.

The idea of uncertainty penalization has been explored as an alternative to empirical risk minimization in statistical learning, starting with Maurer and Pontil [18]. Swaminathan and Joachims [21] applied uncertainty penalization to the offline bandit setting. Their setting is similar to the one we study. An agent seeks to minimize the prediction error of his/her decision, but only observes the loss associated with the selected decision. They assumed that the policy used in the training data is known, which allowed them to use inverse propensity weighting methods. In contrast, we assume ignorability, but not knowledge of the historical policy, and we allow for more complex decision spaces.

We note that our approach bears a superficial resemblance to the upper confidence bound (UCB) algorithms for multi-armed bandits (cf. Bubeck et al. [8]). These algorithms choose the action with the highest upper confidence bound on its predicted expected reward. Our approach, in contrast, chooses the action with the highest lower confidence bound on its predicted expected reward (or lowest upper confidence bound on predicted expected cost). The difference is that UCB algorithms choose actions with high upside to balance exploration and exploitation in the online bandit setting, whereas we work in the offline setting with a focus on solely exploitation.

### 1.3 Contributions

Our primary contribution is an algorithmic framework for observational data driven optimization that allows the decision variable to take values on continuous and multidimensional sets. We consider applications in personalized medicine, in which the decision is the dose of Warfarin to prescribe to a patient, and in pricing, in which the action is the list of prices for several products in a store.

## 2 Approach

In this section, we introduce the uncertainty penalization approach for optimization with observational data. Recall that the observational data consists of $n$ i.i.d. observations, $(X_1, Z_1, Y_1), \ldots, (X_n, Z_n, Y_n)$. For observation $i$, $X_i$ represents the pertinent auxiliary covariates, $Z_i$ is the decision that was applied, and $Y_i$ is the observed response. The first step of the approach is to train a predictive machine learning model to estimate $\mathbb{E}[c(z; Y)|X = x, Z = z]$. When training the predictive model, the feature space is the cartesian product of the auxiliary covariate space and the decision space, $\mathcal{X} \times \mathcal{Z}$. We have several options for how to train the predictive model. We can train the model to predict $Y$, the cost $c(Z, Y)$, or a combination of these two responses. In general, we denote the prediction of the ML algorithm as a linear combination of the cost function evaluated at the training examples,

$$\hat{\mu}(x, z) := \sum_{i=1}^{n} w_i(x, z) c(z; Y_i).$$

We require the predictive model to satisfy a generalization of the honesty property of Wager and Athey [23].

**Assumption 2** (Honesty). *The model trained on $(X_1, Z_1, Y_1), \ldots, (X_n, Z_n, Y_n)$ is honest, i.e., the weights, $w_i(x, z)$, are determined independently of the outcomes, $Y_1, \ldots, Y_n$.*

This honesty assumption reduces the bias of the predictions of the cost. We also enforce several restrictions on the weight functions.

**Assumption 3** (Weights). *For all $(x,z) \in \mathcal{X} \times \mathcal{Z}$, $\sum_{i=1}^{n} w_i(x,z) = 1$ and for all $i$, $w_i(x,z) \in [0, 1/\gamma_n]$. In addition, $\mathcal{X} \times \mathcal{Z}$ can be partitioned into $\Gamma_n$ regions such that if $(x,z)$ and $(x,z')$ are in the same region, $||w(x,z) - w(x,z')||_1 \leq \alpha ||z - z'||_2$.*

The direct approach to solving (1) amounts to choosing $z \in \mathcal{Z}$ that minimizes $\hat{\mu}(x,z)$, for each new instance of auxiliary covariates, $x$. However, the variance of the predicted cost, $\hat{\mu}(x,z)$, can vary with the decision variable, $z$. Especially with a small training sample size, the direct approach, minimizing $\hat{\mu}(x,z)$, can give a decision with a small, but highly uncertain, predicted cost. We can reduce the expected regret of our action by adding a penalty term for the variance of the selected decision. If Assumption 2 holds, the conditional variance of $\hat{\mu}(x,z)$ given $(X_1, Z_1), \ldots, (X_n, Z_n)$ is given by

$$V(x,z) := \sum_i w_i^2(x,z) \mathrm{Var}(c(z; Y_i)|X_i, Z_i).$$

In addition, $\hat{\mu}(x,z)$ may not be an unbiased predictor, so we also introduce a term that penalizes the conditional bias of the predicted cost given $(X_1, Z_1), \ldots, (X_n, Z_n)$. Since the true cost is unknown, it is not possible to exactly compute this bias. Instead, we compute an upper bound under a Lipschitz assumption (details in Section 3).

$$B(x,z) := \sum_i w_i(x,z) ||(X_i, Z_i) - (x,z)||_2.$$

Overall, given a new vector of auxiliary covariates, $x \in \mathcal{X}$, our approach makes a decision by solving

$$\min_{z \in \mathcal{Z}} \hat{\mu}(x,z) + \lambda_1 \sqrt{V(x,z)} + \lambda_2 B(x,z), \tag{2}$$

where $\lambda_1$ and $\lambda_2$ are tuning parameters.

As a concrete example, we can use the CART algorithm of Breiman et al. [7] or the optimal regression tree algorithm of Bertsimas and Dunn [2] as the predictive method. These algorithms work by partitioning the training examples into clusters, i.e., the leaves of the tree. For a new observation, a prediction of the response variable is made by averaging the responses of the training examples that are contained in the same leaf.

$$w_i(x,z) = \begin{cases} \frac{1}{N(x,z)}, & (x,z) \in l(x,z), \\ 0, & \text{otherwise,} \end{cases}$$

where $l(x,z)$ denotes the set of training examples that are contained in the same leaf of the tree as $(x,z)$, and $N(x,z) = |l(x,z)|$. The variance term will be small when the leaf has a large number of training examples, and the bias term will be small when the diameter of the leaf is small. Assumption 2 can be satisfied by ignoring the outcomes when selecting the splits or by dividing the training data into two sets, one for making splits and one for making predictions. Assumption 3 is satisfied with $\alpha = 0$ if the minimum number of training samples in each leaf is $\gamma_n$ and the maximum number of leaves in the tree is $\Gamma_n$.

## 2.1 Parameter Tuning

Before proceeding, we note that the variance terms, $\mathrm{Var}(c(z; Y_i) \mid X_i, Z_i)$, are often unknown in practice. In the absence of further knowledge, we assume homoscedasticity, i.e., $\mathrm{Var}(Y_i|X_i, Z_i)$ is constant. It is possible to estimate this value by training a machine learning model to predict $Y_i$ as a function of $(X_i, Z_i)$ and computing the mean squared error on the training set. However, it may be advantageous to include this value with the tuning parameter $\lambda_1$.

We have several options for tuning parameters $\lambda_1$ and $\lambda_2$ (and whatever other parameters are associated with the predictive model). Because the counterfactual outcomes are unknown, it is not possible to use the standard approach of holding out a validation set during training and evaluating the error of the model on that validation set for each combination of possible parameters. One option is to tune the predictive model's parameters using cross validation to maximize predictive accuracy and then select $\lambda_1$ and $\lambda_2$ using the theory we present in Section 3. Another option is to split the data into a training and validation set and train a predictive model on the validation data to impute the counterfactual outcomes. We then select the model that minimizes the predicted cost on the validation set. For the examples in Section 4, we use a combination of these two ideas. We train a random

forest model on the validation set (in order to impute counterfactual outcomes), and we then select the model that minimizes the sum of the mean squared error and the predicted cost on the validation data. In the supplementary materials, we include computations that demonstrate, for the Warfarin example of Section 4.2, the method is not too sensitive to the choice of $\lambda_1$ and $\lambda_2$.

## 3 Theory

In this section, we describe the theoretical motivation for our approach and provide finite-sample generalization and regret bounds. For notational convenience, we define

$$\mu(x, z) := \mathbb{E}[c(z; Y(z))|X = x] = \mathbb{E}[c(z; Y)|X = x, Z = z],$$

where the second equality follows from the ignorability assumption. Before presenting the results, we first present a few additional assumptions.

**Assumption 4** (Regularity). *The set $\mathcal{X} \times \mathcal{Z}$ is nonempty, closed, and bounded with diameter $D$.*

**Assumption 5** (Objective Conditions). *The objective function satisfies the following properties:*

1. *$|c(z; y)| \leq 1 \quad \forall z, y$.*

2. *For all $y \in \mathcal{Y}$, $c(\cdot; y)$ is L-Lipschitz.*

3. *For any $x, x' \in \mathcal{X}$ and any $z, z' \in \mathcal{Z}$, $|\mu(x, z) - \mu(x', z')| \leq L||(x, z) - (x', z')||$.*

These assumptions provide some conditions under which the generalization and regret bounds hold, but similar results hold under alternative sets of assumptions (e.g. if $c(z; Y)|Z$ is subexponential instead of bounded). With these additional assumptions, we have the following generalization bound. All proofs are contained in the supplementary materials.

**Theorem 1.** *Suppose assumptions 1-5 hold. Then, with probability at least $1 - \delta$,*

$$\mu(x, z) - \hat{\mu}(x, z) \leq \frac{4}{3\gamma_n} \ln(K_n/\delta) + 2\sqrt{V(x, z) \ln(K_n/\delta)} + L \cdot B(x, z) \quad \forall z \in \mathcal{Z},$$

*where $K_n = \Gamma_n \left(9D\gamma_n \left(\alpha(LD + 1 + \sqrt{2}) + L(\sqrt{2} + 3)\right)\right)^p$.*

This result uniformly bounds, with high probability, the true cost of action $z$ by the predicted cost, $\hat{\mu}(x, z)$, a term depending on the uncertainty of that predicted cost, $V(x, z)$, and a term proportional to the bias associated with that predicted cost, $B(x, z)$. It is easy to see how this result motivates the approach described in (2). One can also verify that the generalization bound still holds if $(X_1, Z_1), \ldots, (X_n, Z_n)$ are chosen deterministically, as long as $Y_1, \ldots, Y_n$ are still independent. Using Theorem 1, we are able to derive a finite-sample regret bound.

**Theorem 2.** *Suppose assumptions 1-5 hold. Define*

$$z^* \in \arg\min_z \mu(x, z),$$

$$\hat{z} \in \arg\min_z \hat{\mu}(x, z) + \lambda_1 \sqrt{V(x, z)} + \lambda_2 B(x, z).$$

*If $\lambda_1 = 2\sqrt{\ln(2K_n/\delta)}$ and $\lambda_2 = L$, then with probability at least $1 - \delta$,*

$$\mu(x, \hat{z}) - \mu(x, z^*) \leq \frac{2}{\gamma_n} \ln(2K_n/\delta) + 4\sqrt{V(x, z^*) \ln(2K_n/\delta)} + 2L \cdot B(x, z^*),$$

*where $K_n = \Gamma_n \left(9D\gamma_n \left(\alpha(LD + 1 + \sqrt{2}) + L(\sqrt{2} + 3)\right)\right)^p$.*

By this result, the regret of the approach defined in (2) depends only on the variance and bias terms of the optimal action, $z^*$. Because the predicted cost is penalized by $V(x, z)$ and $B(x, z)$, it does not matter how poor the prediction of cost is at suboptimal actions. Theorem 2 immediately implies the following asymptotic result, assuming the auxiliary feature space and decision space are fixed as the training sample size grows to infinity.

**Corollary 1.** *In the setting of Theorem 2, if $\gamma_n = \Omega(n^\beta)$ for some $\beta > 0$, $\Gamma_n = O(n)$, and $B(x, z^*) \to_p 0$ as $n \to \infty$, then*

$$\mu(x, \hat{z}) \to_p \mu(x, z^*)$$

*as $n \to \infty$.*

The assumptions can be satisfied, for example, with CART or random forest as the learning algorithm with parameters set in accordance with Lemma 2 of Wager and Athey [23]. This next example demonstrates that there exist problems for which the regret of the uncertainty penalized method is strictly better, asymptotically, than the regret of predicted cost minimization.

**Example 1.** *Suppose there are $m + 1$ different actions and two possible, equally probable states of the world. In one state, action 0 has a cost that is deterministically 1, and all other actions have a random cost that is drawn from $\mathcal{N}(0, 1)$ distribution. In the other state, action 0 has a cost that is deterministically 0, and all other actions have a random cost, drawn from a $\mathcal{N}(1, 1)$ distribution. Suppose the training data consists of $m$ trials of each action. If $\hat{\mu}(j)$ is the empirical average cost of action $j$, then the predicted cost minimization algorithm selects the action that minimizes $\hat{\mu}(j)$. The uncertainty penalization algorithm adds a penalty of the form suggested by Theorem 2, $\lambda\sqrt{\frac{\sigma_j^2 \ln m}{m}}$. If $\lambda \geq \sqrt{2}$, the (Bayesian) expected regret of the uncertainty penalization algorithm is asymptotically strictly less than the expected regret of the predicted cost minimization algorithm, $\mathbb{E}R^{UP} = o(\mathbb{E}R^{PCM})$, where the expectations are taken over both the training data and the unknown state of the world.*

This example is simple but demonstrates that there exist settings in which predicted cost minimization is asymptotically suboptimal to the method we have described. In addition, the proof illustrates how one can construct tighter regret bounds than the one in Theorem 2 for problems with specific structure.

## 3.1 Tractability

The tractability of (2) depends on the algorithm that is used as the predictive model. For many kernel-based methods, the resulting optimization problems are highly nonlinear and do not scale well when the dimension of the decision space is more than 2 or 3. For this reason, we advocate using tree-based and linear models as the predictive model. Tree based models partition the space $\mathcal{X} \times \mathcal{Z}$ into $\Gamma_n$ leaves, so there are only $\Gamma_n$ possible values of $w(x, z)$. Therefore, we can solve (2) separately for each leaf. For $j = 1, \ldots, \Gamma_n$, we solve

$$
\begin{aligned}
\min \quad & \hat{\mu}(x, z) + \lambda_1 \sqrt{V(x, z)} + \lambda_2 B(x, z) \\
\text{s.t.} \quad & z \in \mathcal{Z} \\
& (x, z) \in L_j,
\end{aligned}
\tag{3}
$$

where $L_j$ denotes the subset of $\mathcal{X} \times \mathcal{Z}$ that makes up leaf $j$ of the tree. Because each split in the tree is a hyperplane, $L_j$ is defined by an intersection of hyperplanes and thus is a polyhedral set. Clearly, $B(x, z)$ is a convex function in $z$, as it is a nonnegative linear combination of convex functions. If we assume homoscedasticity, then $V(x, z)$ is constant for all $(x, z) \in L_j$. If $c(z; y)$ is convex in $z$ and $\mathcal{Z}$ is a convex set, (3) is a convex optimization problem and can be solved by convex optimization techniques. Furthermore, since the $\Gamma_n$ instances of (3) are all independent, we can solve them in parallel. Once (3) has been solved for all leaves, we select the solution from the leaf with the overall minimal objective value.

For tree ensemble methods, such as random forest [6] or xgboost [9], optimization is more difficult. We compute optimal decisions using a coordinate descent heuristic. From a random starting action, we cycle through holding all decision variables fixed except for one and optimize that decision using discretization. We repeat this until convergence from several different random starting decisions. For linear predictive models, the resulting problem is often a second order conic optimization problem, which can be handled by off-the-shelf solvers (details given in the supplementary materials).

# 4 Results

In this section, we demonstrate the effectiveness of our approach with two examples. In the first, we consider pricing problem with synthetic data, while in the second, we use real patient data for personalized Warfarin dosing.

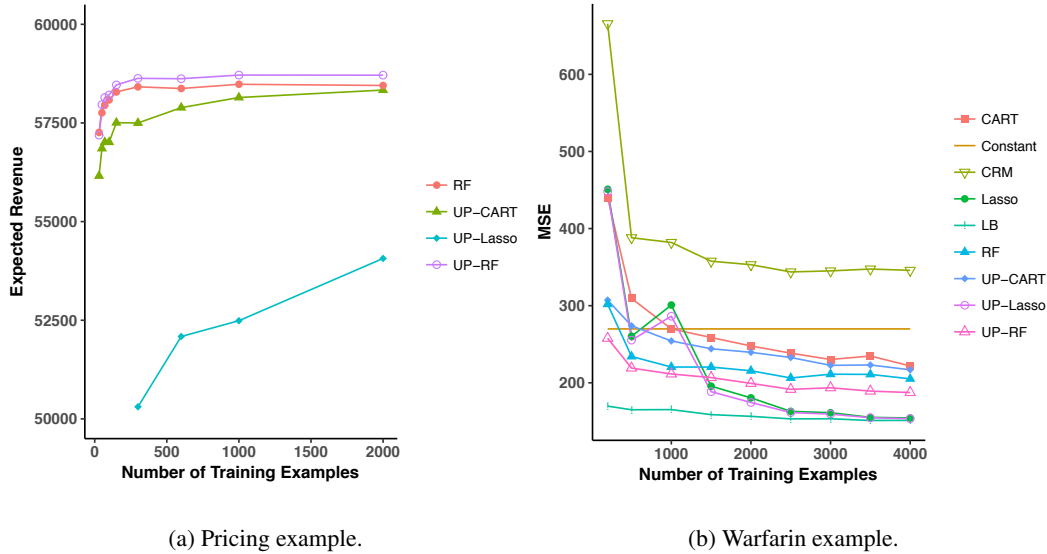

(a) Pricing example.

(b) Warfarin example.

Figure 1

## 4.1 Pricing

In this example, the decision variable, $z \in \mathbb{R}^5$, is a vector of prices for a collection of products. The outcome, $Y$, is a vector of demands for those products. The auxiliary covariates may contain data on the weather and other exogenous factors that may affect demand. The objective is to select prices to maximize revenue for a given vector of auxiliary covariates. The demand for a single product is affected by the auxiliary covariates, the price of that product, and the price of one or more of the other products, but the mapping is unknown to the algorithm. The details on the data generation process can be found in the supplementary materials.

In Figure 1a, we compare the expected revenues of the strategies produced by several algorithms. CART, RF, and Lasso refer to the direct methods of training, respectively, a decision tree, a random forest, and a lasso regression [22] to predict revenue, as a function of the auxiliary covariates and prices, and choosing prices, for each vector of auxiliary covariates in the test set, that maximize predicted revenue. (Note that the revenues for CART and Lasso were too small to be displayed on the plot. Unsurprisingly, the linear model performs poorly because revenue does not vary linearly with price. We restrict all prices to be at most 50 to ensure the optimization problems are bounded.) UP-CART, UP-RF, and UP-Lasso refer to the uncertainty penalized analogues in which the variance and bias terms are included in the objective. For each training sample size, $n$, we average our results over one hundred separate training sets of size $n$. At a training size of 2000, the uncertainty penalized random forest method improves expected revenue by an average of \$270 compared to the direct RF method. This improvement is statistically significant at the 0.05 significance level by the Wilcoxon signed-rank test ($p$-value $4.4 \times 10^{-18}$, testing the null hypothesis that mean improvement is 0 across 100 different training sets).

## 4.2 Warfarin Dosing

Warfarin is a commonly prescribed anticoagulant that is used to treat patients who have had blood clots or who have a high risk of stroke. Determining the optimal maintenance dose of Warfarin presents a challenge as the appropriate dose varies significantly from patient to patient and is potentially affected by many factors including age, gender, weight, health history, and genetics. However, this is a crucial task because a dose that is too low or too high can put the patient at risk for clotting or bleeding. The effect of a Warfarin dose on a patient is measured by the International Normalilzed Ratio (INR). Physicians typically aim for patients to have an INR in a target range of 2-3.

In this example, we test the efficacy of our approach in learning optimal Warfarin dosing with data from Consortium et al. [10]. This publicly available data set contains the optimal stable dose, found

by experimentation, for a diverse set of 5410 patients. In addition, the data set contains a variety of covariates for each patient, including demographic information, reason for treatment, medical history, current medications, and the genotype variant at CYP2C9 and VKORC1. It is unique because it contains the optimal dose for each patient, permitting the use of off-the-shelf machine learning methods to predict this optimal dose as a function of patient covariates. We instead use this data to construct a problem with observational data, which resembles the common problem practitioners face. Our access to the true optimal dose for each patient allows us to evaluate the performance of our method out-of-sample. This is a commonly used technique, and the resulting data set is sometimes called *semi-synthetic*. Several researchers have used the Warfarin data for developing personalized approaches to medical treatments. In particular, Kallus [15] and Bertsimas et al. [3] tested algorithms that learned to treat patients from semi-synthetic observational data. However, they both discretized the dosage into three categories, whereas we treat the dosage as a continuous decision variable.

To begin, we split the data into a training set of 4000 patients and a test set of 1410 patients. We keep this split fixed throughout all of our experiments to prevent cheating by using insights gained by visualization and exploration on the training set. Similar to Kallus [15], we assume physicians prescribe Warfarin as a function of BMI. We assume the response that the physicians observe is related to the difference between the dose a patient was given and the true optimal dose for that patient. It is a noisy observation, but it, on average, gives directional information (whether the dose was too high or too low) and information on the magnitude of the distance from the optimal dose. The precise details of how we generate the data are given in the supplementary materials. For all methods, we repeat our work across 100 randomizations of assigned training doses and responses. To measure the performance of our methods, we compute, on the test set, the mean squared error (MSE) of the prescribed doses relative to the true optimal doses. Using the notation described in Section 1, $X_i \in \mathbb{R}^{99}$ represents the auxiliary covariates for patient $i$. We work in normalized units so the covariates all contribute equally to the bias penalty term. $Z_i \in \mathbb{R}$ represents the assigned dose for patient $i$, and $Y_i \in \mathbb{R}$ represents the observed response for patient $i$. The objective in this problem is to minimize $(\mathbb{E}[Y(z)|X = x])^2$ with respect to the dose, $z$.[1]

Figure 1b displays the results of several algorithms as a function of the number of training examples. We compare CART, without any penalization, to CART with uncertainty penalization (UP-CART), and we see that uncertainty penalization offers a consistent improvement. This improvement is greatest when the training sample size is smallest. (Note: for CART with no penalization, when multiple doses give the same optimal predicted response, we select the mean.) Similarly, when we compare the random forest and Lasso methods with their uncertainty-penalizing analogues, we again see consistent improvements in MSE. The "Constant" line in the plot measures the performance of a baseline heuristic that assigns a fixed dose of 35 mg/week to all patients. The "LB" line provides an unattainable lower bound on the performance of all methods that use the observational data. For this method, we train a random forest to predict the optimal dose as a function of the patient covariates. We also compare our methods with the Counterfactual Risk Minimization (CRM) method of Swaminathan and Joachims [21]. We allow their method access to the true propensity scores that generated the data and optimize over all regularized linear policies for which the proposed dose is a linear function of the auxiliary covariates. We tried multiple combinations of tuning parameters, but the method always performed poorly out-of-sample. We suspect this is due to the size of the policy space. Our lasso based method works best on this data set when the number of training samples is large, but the random forest based method is best for smaller sample sizes. With the maximal training set size of 4000, the improvements of the CART, random forest, and lasso uncertainty penalized methods over their unpenalized analogues (2.2%, 8.6%, 0.5% respectively) are all statistically significant at the 0.05 family-wise error rate level by the Wilcoxon signed-rank test with Bonferroni correction (adjusted $p$-values $2.1 \times 10^{-4}, 4.3 \times 10^{-16}, 1.2 \times 10^{-6}$ respectively).

## 5 Conclusions

In this paper, we introduced a data-driven framework that combines ideas from predictive machine learning and causal inference to optimize an uncertain objective using observational data. Unlike

most existing algorithms, our approach handles continuous and multi-dimensional decision variables by introducing terms that penalize the uncertainty associated with the predicted costs. We proved finite sample generalization and regret bounds and provided a sufficient set of conditions under which the resulting decisions are asymptotically optimal. We demonstrated, both theoretically and with real-world examples, the tractability of the approach and the benefit of the approach over unpenalized predicted cost minimization.

## Footnotes

[1]This objective differs slightly from the setting described in Section 3 in which the objective was to minimize the conditional expectation of a cost function. However, it is straightforward to modify the results to obtain the same regret bound (save a few constant factors) when minimizing $g(\mathbb{E}[c(z; Y(z))|X = x])$ for a Lipschitz function, $g$.

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
