[Supplementary Material]

# Appendix: Optimization over Continuous and Multi-dimensional Decisions with Observational Data

## A   Proofs

To begin, we prove the following lemma.

**Lemma 1.** *Suppose assumptions 1-5 hold. If $(x, z)$ and $(x, z')$ are in the same partition of $\mathcal{X} \times \mathcal{Z}$, as specified by assumption 3, then*

$$|\Psi(z, \delta) - \Psi(z', \delta)| \leq \left( \alpha(LD + 1 + \sqrt{2\lambda_{\max} \ln 1/\delta}) + L(\sqrt{2 \ln 1/\delta} + 3) \right) ||z - z'||,$$

*where $\Psi(z, \delta) = \mu(x, z) - \hat{\mu}(x, z) - \frac{2}{3\gamma_n} \ln(1/\delta) - \sqrt{2V(x, z) \ln(1/\delta)} - L \cdot B(x, z).$*

*Proof.* We first note $|\mu(x, z) - \mu(x, z')| \leq L||z - z'||$ by the Lipschitz assumption on $c(z; y)$.
   Next, since $(x, z)$ and $(x, z')$ are contained in the same partition,

$$
\begin{aligned}
|\hat{\mu}(x, z) - \hat{\mu}(x, z')| &= \left| \sum_i w_i(x, z)c(z; Y_i) - w_i(x, z')c(z'; Y_i) \right| \\
&\leq \left| \sum_i w_i(x, z)c(z; Y_i) - w_i(x, z)c(z'; Y_i) \right| \\
&\quad + \left| \sum_i w_i(x, z)c(z'; Y_i) - w_i(x, z')c(z'; Y_i) \right| \\
&\leq L||z - z'|| + ||w(x, z) - w(x, z')||_1 \\
&\leq (L + \alpha)||z - z'||,
\end{aligned}
$$

where we have used Holder's inequality, the uniform bound on $c$, and Assumption 3.

Similarly, for the bias term,

$$
\begin{aligned}
|LB(x,z) - LB(x,z')| &= L\left|\sum_i w_i(x,z)||(X_i,Z_i) - (x,z)|| - w_i(x,z')||(X_i,Z_i) - (x,z')||\right| \\
&\leq L\left|\sum_i w_i(x,z)||(X_i,Z_i) - (x,z)|| - w_i(x,z)||(X_i,Z_i) - (x,z')||\right| \\
&\quad + L\left|\sum_i w_i(x,z)||(X_i,Z_i) - (x,z')|| - w_i(x,z')||(X_i,Z_i) - (x,z')||\right| \\
&\leq L\sum_i w_i(x,z)\left|||(X_i,Z_i) - (x,z)|| - ||(X_i,Z_i) - (x,z')||\right| \\
&\quad + L||w(x,z) - w(x,z')||_1 \sup_i ||(X_i,Z_i) - (x,z)|| \\
&\leq (L + L\alpha D)||z - z'||.
\end{aligned}
$$

Next, we consider variance term. We let $\Sigma(z)$ denote the diagonal matrix with $\mathrm{Var}(c(z;Y_i)|X_i,Z_i)$ for $i = 1,\ldots,n$ as entries. As before,

$$
\begin{aligned}
|\sqrt{V(x,z)} - \sqrt{V(x,z')}| &= \left|\sqrt{\sum_i w_i^2(x,z)\mathrm{Var}(c(z;Y_i)|X_i,Z_i)} - \sqrt{\sum_i w_i^2(x,z')\mathrm{Var}(c(z';Y_i)|X_i,Z_i)}\right| \\
&\leq \left|\sqrt{\sum_i w_i^2(x,z)\mathrm{Var}(c(z;Y_i)|X_i,Z_i)} - \sqrt{\sum_i w_i^2(x,z)\mathrm{Var}(c(z';Y_i)|X_i,Z_i)}\right| \\
&\quad + \left|\sqrt{\sum_i w_i^2(x,z)\mathrm{Var}(c(z';Y_i)|X_i,Z_i)} - \sqrt{\sum_i w_i^2(x,z')\mathrm{Var}(c(z';Y_i)|X_i,Z_i)}\right| \\
&= \left|\sqrt{w(x,z)^T\Sigma(z)w(x,z)} - \sqrt{w(x,z)^T\Sigma(z')w(x,z)}\right| \\
&\quad + \left|||w(x,z)||_{\Sigma(z')} - ||w(x,z')||_{\Sigma(z')}\right|,
\end{aligned}
$$

where $||v||_\Sigma = \sqrt{v^T\Sigma v}$. One can verify that, because $\Sigma$ is positive semidefinite, $||\cdot||_\Sigma$ is seminorm that satisfies the triangle inequality. Therefore, we can upper bound the latter term by

$$
\begin{aligned}
\sqrt{(w(x,z) - w(x,z'))^T\Sigma(w(x,z) - w(x,z'))} &\leq ||w(x,z) - w(x,z')|| \\
&\leq ||w(x,z) - w(x,z')||_1 \\
&\leq \alpha||z - z'||,
\end{aligned}
$$

where we have used the assumption that $|c(z;y)| \leq 1$.

The former term can again be upper bounded by the triangle inequality.

$$\left| \sqrt{\sum_i w_i^2(x,z)\mathrm{Var}(c(z;Y_i)|X_i,Z_i)} - \sqrt{\sum_i w_i^2(x,z)\mathrm{Var}(c(z';Y_i)|X_i,Z_i)} \right|$$

$$\leq \sqrt{\sum_i w_i^2(x,z)(\sqrt{\mathrm{Var}(c(z;Y_i)|X_i,Z_i)} - \sqrt{\mathrm{Var}(c(z';Y_i)|X_i,Z_i)})^2} \qquad (1)$$

Noting that $\sqrt{\mathrm{Var}(c(z;Y_i))} = ||c(z;Y_i) - \mathbb{E}[c(z;Y_i)]||_{L_2}$ (dropping conditioning for notational convenience), we can apply the triangle inequality to the $L_2$ norm:

$$\left( ||c(z;Y_i) - \mathbb{E}[c(z;Y_i)]||_{L_2} - ||c(z';Y_i) - \mathbb{E}[c(z';Y_i)]||_{L_2} \right)^2$$
$$\leq ||c(z;Y_i) - c(z';Y_i) - \mathbb{E}[c(z;Y_i) - c(z';Y_i)]||_{L_2}^2$$
$$\leq \mathbb{E}[(c(z;Y_i) - c(z';Y_i))^2]$$
$$\leq L^2||z - z'||^2.$$

Therefore, we can upperbound (1) by

$$\sqrt{\sum_i w_i^2(x,z)L^2||z-z'||^2}$$

$$\leq \sum_i w_i(x,z)L||z-z'|| = L||z-z'||,$$

where we have used the concavity of the square root function. Therefore,

$$|\sqrt{V(x,z)} - \sqrt{V(x,z')}| \leq (\alpha + L)||z - z'||.$$

Combining the three results with the triangle inequality yields the desired result. □

*Proof of Theorem 1.* To derive a regret bound, we first restrict our attention to the fixed design setting. Here, we condition on $X_1, Z_1, \ldots, X_n, Z_n$ and bound $\hat{\mu}(x,z)$ around its expectation. To simplify notation, we write $X$ to denote $(X_1, \ldots, X_n)$ and $Z$ to denote $(Z_1, \ldots, Z_n)$. Note that by the honesty assumption, in this setting, $\hat{\mu}$ is a simple sum of independent random variables. Applying Bernstein's inequality (see, for example, Boucheron et al. [1]), we have, for $\delta \in (0,1)$,

$$P\left( \mathbb{E}[\hat{\mu}(x,z) \mid X, Z] - \hat{\mu}(x,z) \leq \frac{2}{3\gamma_n}\ln(1/\delta) + \sqrt{2V(x,z)\ln(1/\delta)} \,\middle|\, X, Z \right) \geq 1 - \delta.$$

Next, we need to bound the difference between $\mathbb{E}[\hat{\mu}(x,z)|X,Z]$ and $\mu(x,z)$. By the honesty assumption, Jensen's inequality, and the Lipschitz assumption, we have

$$|\mathbb{E}[\hat{\mu}(x,z) \mid X, Z] - \mu(x,z)| = \left| \sum_i w_i(x,z)(\mu(X_i,Z_i) - \mu(x,z)) \right|$$

$$\leq \sum_i w_i(x,z)|\mu(X_i,Z_i) - \mu(x,z)|$$

$$\leq L \sum_i w_i(x,z)||(X_i,Z_i) - (x,z)||$$

$$= L \cdot B(x,z).$$

Combining this with the previous result, we have, with probability at least $1-\delta$ (conditioned on $X$ and $Z$),

$$\mu(x,z) - \hat{\mu}(x,z) \leq \frac{2}{3\gamma_n}\ln(1/\delta) + \sqrt{2V(x,z)\ln(1/\delta)} + L \cdot B(x,z) \qquad (2)$$

Next, we extend this result to hold uniformly over all $z \in \mathcal{Z}$. To do so, we partition $\mathcal{X} \times \mathcal{Z}$ into $\Gamma_n$ regions as in Assumption 3. For each region, we construct a $\nu$-net. Therefore, we have a set $\{\hat{z}_1, \ldots, \hat{z}_{K_n}\}$ such that for any $z \in \mathcal{Z}$, there exists a $\hat{z}_k$ such that $(x,z)$ and $(x, \hat{z}_k)$ are contained in the same region with $||z - \hat{z}_k|| \leq \nu$. For ease of notation, let $k : \mathcal{Z} \rightarrow \{1, \ldots, K_n\}$ return an index that satisfies these criteria. By assumption, $\mathcal{Z} \subset \mathbb{R}^p$ has finite diameter, $D$, so we can construct this set with $K_n \leq \Gamma_n(3D/\nu)^p$ (e.g., Shalev-Shwartz and Ben-David [3, pg. 337]).

By Lemma 1 (and using the notation therein), we have

$$\Psi(z,\delta) \leq \Psi(\hat{z}_{k(z)}, \delta) + \nu\left(\alpha(LD + 1 + \sqrt{2\ln 1/\delta}) + L(\sqrt{2\ln 1/\delta} + 3)\right).$$

Taking the supremum over $z$ of both sides, we get

$$\sup_z \Psi(z,\delta) \leq \max_k \Psi(\hat{z}_k, \delta) + \nu\left(\alpha(LD + 1 + \sqrt{2\ln 1/\delta}) + L(\sqrt{2\ln 1/\delta} + 3)\right).$$

If we let $\nu = \frac{1}{3\gamma_n}\left(\alpha(LD + 1 + \sqrt{2}) + L(\sqrt{2} + 3)\right)^{-1}$, we have

$$P(\sup_z \Psi(z,\delta) > 0|X, Z)$$

$$\leq P\left(\max_k \Psi(\hat{z}_k, \delta) + \nu\left(\alpha(LD + 1 + \sqrt{2\ln 1/\delta}) + L(\sqrt{2\ln 1/\delta} + 3)\right) > 0 \middle| X, Z\right)$$

$$\leq P\left(\max_k \Psi(\hat{z}_k, \delta) + \nu\left(\alpha(LD + 1 + \sqrt{2}) + L(\sqrt{2} + 3)\right)\ln 1/\delta > 0 \middle| X, Z\right)$$

$$\leq \sum_k P\left(\Psi(\hat{z}_k, \delta) + \frac{\ln 1/\delta}{3\gamma_n} > 0 \middle| X, Z\right)$$

$$\leq \sum_k P\left(\Psi(\hat{z}_k, \sqrt{\delta}) > 0 \middle| X, Z\right)$$

$$\leq K_n\sqrt{\delta},$$

where we have used the union bound and (2). Replacing $\delta$ with $\delta^2/K_n^2$ and integrating both sides to remove the conditioning completes the proof. $\square$

*Proof of Theorem 2.* By Theorem 1, with probability at least $1 - \delta/2$,

$$\mu(x, \hat{z}) \leq \hat{\mu}(x, \hat{z}) + \frac{4}{3\gamma_n}\ln(2K_n/\delta) + \lambda_1\sqrt{V(x, \hat{z})} + \lambda_2 B(x, \hat{z})$$

$$\leq \hat{\mu}(x, z^*) + \frac{4}{3\gamma_n}\ln(2K_n/\delta) + \lambda_1\sqrt{V(x, z^*)} + \lambda_2 B(x, z^*),$$

where the second inequality follows from the definition of $\hat{z}$. Using the same argument we used to derive (2), since $z^*$ is not a random quantity, we have, with probability at least $1 - \delta/2$,

$$\hat{\mu}(x, z^*) - \mu(x, z^*) \leq \frac{2}{3\gamma_n} \ln(2/\delta) + \sqrt{2V(x, z^*) \ln(2/\delta)} + L \cdot B(x, z^*)$$

$$\leq \frac{2}{3\gamma_n} \ln(2K_n/\delta) + \lambda_1 \sqrt{V(x, z^*)} + \lambda_2 B(x, z^*).$$

Combining the two inequalities with the union bound yields the desired result. $\qquad\square$

*Proof of Corollary 1.* We show $\mu(x, \hat{z}) - 2LB(x, z^*) \to_p \mu(x, z^*)$. The desired result follows from the assumption regarding $B(x, z^*)$ and Slutsky's theorem. First, we note, due to the assumption $|c(z; y)| \leq 1$,

$$V(x, z^*) = \sum_i w_i(x, z^*)\text{Var}(c(z^*; Y_i)|X_i, Z_i) \leq \frac{1}{\gamma_n} \sum_i w_i(x, z^*) = \frac{1}{\gamma_n}.$$

We have, for any $\epsilon > 0$,

$$P(|\mu(x, \hat{z}) - 2LB(x, z^*) - \mu(x, z^*)| > \epsilon)$$
$$\leq P(\mu(x, \hat{z}) - 2LB(x, z^*) - \mu(x, z^*) > \epsilon/2)$$
$$+ P(\mu(x, z^*) - \mu(x, \hat{z}) + 2LB(x, z^*) > \epsilon/2).$$

By Theorem 2, for large enough $n$, the first term is upper bounded by

$$2K_n \exp\left(-\frac{\epsilon^2}{4(2/\gamma_n + 4\sqrt{V(x, z^*)})^2}\right)$$
$$\leq 2K_n \exp\left(-\frac{\epsilon^2}{4(2/\sqrt{\gamma_n} + 4/\sqrt{\gamma_n})^2}\right)$$
$$= 2\Gamma_n \left(9D\gamma_n \left(\alpha(LD + 1 + \sqrt{2}) + L(\sqrt{2} + 3)\right)\right)^p \exp\left(-\frac{\gamma_n \epsilon^2}{144}\right)$$
$$\leq C_1 n^{1+\beta} \exp(-C_2 n^\beta) \to 0.$$

Because $\mu(x, z^*) \leq \mu(x, \hat{z})$, the latter term is upper bounded by

$$P(B(x, z^*) > \epsilon/4L) \to 0.$$

$\qquad\square$

*Proof of Example 1.* First we consider the case that the zero variance action has cost 0, and the other actions have cost 1 (call this event $A$). Because the cost of the optimal action is 0 and the cost of a suboptimal action is 1, the expected regret in this problem equals the probability of the algorithm selecting a suboptimal action. Noting that $\hat{\mu}(j) \sim \mathcal{N}(1, 1/m)$ for $j = 1, \ldots, m$, we can express the expected regret of the predicted cost minimization algorithm as

$$\mathbb{E}[R^{PCM}|A] = P\left(\hat{\mu}(j) < 0 \text{ for some } j \in \{1, \ldots, m\}|A\right) = P\left(\max_j W_j > \sqrt{m}\right),$$

where $W_1, \ldots, W_m$ are i.i.d. standard normal random variables. Similarly, the expected regret of the uncertainty penalized algorithm can be expressed as

$$\mathbb{E}[R^{UP}|A] = P\left(\hat{\mu}(j) < -\frac{\lambda\sqrt{\ln m}}{\sqrt{m}} \text{ for some } j \in \{1, \ldots, m\} \Big| A\right)$$

$$= P\left(\max_j W_j > \sqrt{m} + \lambda\sqrt{\ln m}\right)$$

We can construct an upper bound on $\mathbb{E}R^{UP}$ with the union bound and a concentration inequality (as in the proof of Theorem 1). Applying the Gaussian tail inequality (see, for example, Vershynin [5, Proposition 2.1.2]), we have

$$\mathbb{E}[R^{UP}|A] \le mP(W_1 > \sqrt{m} + \lambda\sqrt{\ln m})$$

$$\le \frac{\sqrt{m}}{\sqrt{2\pi}}\exp\left(-\frac{1}{2}(\sqrt{m} + \lambda\sqrt{\ln m})^2\right)$$

$$= \frac{\sqrt{m}}{m^{\lambda^2/2}\sqrt{2\pi}}\exp(-m/2)\exp(-\lambda\sqrt{m\ln m})$$

$$\le \frac{1}{\sqrt{m}\sqrt{2\pi}}e^{-m/2},$$

where we have used the assumption $\lambda \ge \sqrt{2}$.

To lower bound the expected regret of the predicted cost minimization algorithm, we can use a similar Gaussian tail inequality.

$$\mathbb{E}[R^{PCM}|A] = 1 - \left[1 - P(W_1 > \sqrt{m})\right]^m$$

$$\ge 1 - \left[1 - \left(1 - \frac{1}{m}\right)\frac{1}{\sqrt{m}\sqrt{2\pi}}e^{-m/2}\right]^m$$

$$\ge 1 - \left[1 - \frac{1}{2\sqrt{m}\sqrt{2\pi}}e^{-m/2}\right]^m$$

$$\ge 1 - \left[\left[1 - \frac{1}{2\sqrt{m}\sqrt{2\pi}}e^{-m/2}\right]^{2\sqrt{2\pi m}\exp(m/2)}\right]^{\sqrt{m}\exp(-m/2)/2\sqrt{2\pi}},$$

where the second inequality is valid for all $m \ge 2$. One can verify that $(1 - 1/n)^n$ is a monotonically increasing function that converges to $e^{-1}$. Therefore, for all $m \ge 2$,

$$\mathbb{E}[R^{PCM}|A] \ge 1 - \exp\left(-\frac{\sqrt{m}}{2\sqrt{2\pi}}\exp(-m/2)\right).$$

Next, we use these bounds to compute the ratio $\mathbb{E}[R^{UP}|A]/\mathbb{E}[R^{PCM}|A]$ in the limit as $m \to \infty$.

$$\frac{\mathbb{E}[R^{UP}|A]}{\mathbb{E}[R^{PCM}|A]} \le \frac{\frac{1}{\sqrt{m}\sqrt{2\pi}}e^{-m/2}}{1 - \exp\left(-\frac{\sqrt{m}}{2\sqrt{2\pi}}\exp(-m/2)\right)}.$$

Applying L'Hopital's rule, the limit of the right hand side is equal to the limit of

$$\frac{2(2\pi)^{-1/2}\left(-m^{-3/2}e^{-m/2}-m^{-1/2}e^{-m/2}\right)}{(2\pi)^{-1/2}\left[m^{-1/2}e^{-m/2}-m^{1/2}e^{-m/2}\right]\exp\left(-\frac{\sqrt{m}}{2\sqrt{2\pi}}e^{-m/2}\right)}$$

$$=2\frac{-1-m}{m-m^2}\cdot\exp\left(\frac{\sqrt{m}}{2\sqrt{2\pi}}e^{-m/2}\right)\to 0.$$

Next, we consider the case that the zero variance action has cost 1, and the other actions have cost 0. The expected regret equals the probability that the zero variance action is selected. For sufficiently large $m$,

$$\mathbb{E}[R^{UP}|A^c]=P\left(\hat{\mu}(j)>1-\frac{\lambda\sqrt{\ln m}}{\sqrt{m}}\quad\forall j\in\{1,\ldots,m\}\middle|A^c\right)$$

$$\leq P(W_1>\sqrt{m}-\lambda\sqrt{\ln m})^m$$

$$\leq P(W_1>\sqrt{m}/2)^m$$

$$\leq\left(\frac{2}{\sqrt{2\pi}}e^{-m/8}\right)^m$$

$$\leq e^{-m^2/8}=o(\mathbb{E}[R^{UP}|A]).$$

Therefore, for sufficiently large $m$ and some constant $C$,

$$\frac{\mathbb{E}[R^{UP}]}{\mathbb{E}[R^{PCM}]}=\frac{\mathbb{E}[R^{UP}|A]+\mathbb{E}[R^{UP}|A^c]}{\mathbb{E}[R^{PCM}|A]+\mathbb{E}[R^{PCM}|A^c]}$$

$$\leq\frac{\mathbb{E}[R^{UP}|A]+\mathbb{E}[R^{UP}|A^c]}{\mathbb{E}[R^{PCM}|A]}$$

$$\leq(1+C)\frac{\mathbb{E}[R^{UP}|A]}{\mathbb{E}[R^{PCM}|A]}\to 0.$$

$\square$

# B  Optimization with Linear Predictive Models

Here, we detail the optimization of (2) with linear predictive models. We focus on the case that $c(z;Y)=Y$ for simplicity. For these models, we posit the outcome is a linear function of the auxiliary covariates and decision. That is there exists a $\beta$ such that, given $X=x$, $Y(z)=(x,z)^T\beta+\epsilon$, where $\epsilon$ is a mean 0 subgaussian noise term with variance $\sigma^2$. If we let $A$ denote the design matrix for the problem, a matrix with rows consisting of $(X_i,Z_i)$ for $i=1,\ldots,n$, then the ordinary least squares (OLS) estimator for $\beta$ is given by

$$\hat{\beta}^{OLS}=(A^TA)^{-1}A^TY.$$

The ordinary least squares estimator is unbiased, so when solving (2), we set $\lambda_2=0$. The variance of $(x,z)^T\hat{\beta}^{OLS}$ is given by $\sigma^2(x,z)^T(A^TA)^{-1}(x,z)$. $(A^TA)^{-1}$ is a positive semidefinite matrix, so $\sqrt{V(x,z)}$ is convex. Therefore, (2) becomes

$$\min_{z\in\mathcal{Z}}\ (x,z)^T\hat{\beta}^{OLS}+\lambda_1\sigma\sqrt{(x,z)^T(A^TA)^{-1}(x,z)},$$

which is a second order conic optimization problem if $\mathcal{Z}$ is polyhedral and can be solved efficiently by commercial solvers. Even if $\mathcal{Z}$ is a mixed integer set, commercial solvers such as Gurobi [2] can still solve the problem for sizes of practical interest.

For regularized linear models such as ridge and lasso regression, we use a similar approach. Although these estimators are biased, we set $\lambda_2 = 0$ for computational reasons. The ridge estimator for $\beta$ has a similar form to the OLS estimator:

$$\hat{\beta}^{Ridge} = (A^T A + \alpha I)^{-1} A^T Y,$$

for some $\alpha \geq 0$. The resulting optimization problem is essentially the same as with the OLS estimator. The lasso estimator does not have a closed form solution, but we can approximate it as in Tibshirani [4]:

$$P\hat{\beta}^{Lasso} \approx (PA^T AP^T + \alpha PW)^{-1} PA^T Y,$$

where $W = \mathrm{diag}(1/|\beta_1^*|, \ldots, 1/|\beta_{d+p}^*|)$, $\beta^*$ is the true lasso solution, and $P$ is a projection matrix that projects to the nonzero components of $\beta^*$. (The zero components of $\beta^*$ are still 0 in the approximation.) With this approximation, the resulting optimization takes the same form as those for the OLS and ridge estimators.

# C   Data Generation

## C.1   Pricing

For our synthetic pricing example, we consider a store offering 5 products. We generate auxiliary covariates, $X_i$, from a $\mathcal{N}(10, 1)$ distribution. We generate historical prices, $Z_i$, from a Gaussian distribution,

$$\mathcal{N}\left( X_i^T \begin{pmatrix} 1 & 0 \\ 1 & 0 \\ 0 & 1 \\ 0 & 1 \\ 0.5 & 0.5 \end{pmatrix}, 100I \right).$$

We compute the expected demand for each product as:

$$\mu = \begin{pmatrix} 500 - (Z_i^1)^2/10 - X_i^1 \cdot Z_i^1/10 - (X_i^1)^2/10 - Z_i^2 \\ 500 - (Z_i^2)^2/10 - X_i^1 \cdot Z_i^2/10 - (X_i^1)^2/10 - Z_i^1 \\ 500 - (Z_i^3)^2/10 - X_i^2 \cdot Z_i^3/10 - (X_i^2)^2/10 + Z_i^1 + Z_i^2 \\ 500 - (Z_i^4)^2/10 - X_i^2 \cdot Z_i^4/10 - (X_i^2)^2/10 + Z_i^1 + Z_i^2 \\ 500 - (Z_i^5)^2/10 - X_i^2 \cdot Z_i^5/20 - X_i^1 \cdot Z_i^5/20 - (X_i^2)^2/10 \end{pmatrix},$$

and generate $Y_i$ from a $\mathcal{N}(\mu, 2500I)$ distribution. This example serves to simulate the situation in which some products are complements and some are substitutes.

## C.2    Warfarin Dosing

To simulate how physicians might assign Warfarin doses to patients, we compute a normalized BMI for each patient (i.e. body mass divided by height squared, normalized by the population standard deviation of BMI). For each patient, we then sample a dose (in mg/week), $Z_i$, from

$$Z_i \sim \mathcal{N}(30 + 15 \cdot \mathrm{BMI}_i, 64).$$

If $Z_i$ is negative, we assign a dose drawn uniformly from $[0, 20]$. If the data dose not contain the patients height and/or weight, we assign a dose drawn uniformly from $[10, 50]$, a standard range for Warfarin doses.

   To simulate the response that a physician observes for a particular patient, we compute the difference between the the assigned dose and the true optimal dose for that patient, $Z_i^*$, and add noise. We then cap the response so it is less than or equal to 40 in absolute value. The reasoning behind this construction is that the INR measurement gives the physician some idea of whether the assigned dose is too high or too low and whether it is close to the optimal dose. However, if the dose is very far from optimal, then the information INR provides is not very useful in determining the optimal dose (it is purely directional). The response of patient $i$ is given by

$$Y_i = \begin{cases} -40, & R_i < -40 \\ R_i, & -40 \leq R_i \leq 40 \\ 40, & R_i > 40 \end{cases},$$

where $R_i \sim \mathcal{N}(Z_i - Z_i^*, 400)$.

# D    Sensitivity to Selection of Tuning Parameters

To test the sensitivity of the method to the selection of tuning parameters, we conduct an experiment on the Warfarin example with the random forest as the base learner. We compute the out-of-sample error for many combinations of $\lambda_1$ and $\lambda_2$. From Figure **??**, we see that the out-of-sample performance is not too sensitive to the selection of parameters. All of the selected parameter combinations out-perform the unpenalized method with the exception of $(\lambda_1 = 100, \lambda_2 = 0)$, which is an extreme choice. This demonstrates that the tuning parameter selection does not have to be extremely precise to improve performance.