[Reviews · NeurIPS 2018]

Reviewer 1



The paper proposes an algorithm that can learn good decision-making policies over a continuous set of decisions using only access to observational data. The problem is well-motivated, but the paper can be substantially strengthened. Quality: Ok The paper clearly motivates a weakness of direct regression (e.g. from context and decision, to predict expected cost). The regression models may have different uncertainty for different decisions, and so it is useful to include an empirical estimate of variance and bias of the regression when selecting decisions. The paper will be more informative by highlighting several choices of regression models, each with different V (variance) and B (bias), and observing how lambda_1 lambda_2 are tuned to pick low-cost decisions with high probability. Clarity: Ok The high level message is clear, but several improvements can be made. E.g. Till Line 123, I assumed that the cost c(z,y) is unknown and is being modeled. Equation for hat{mu} however seems to imply that the costs are known and only the outcome distribution is unknown (so that hat{mu} can be written as a combination of training responses c(z, Y_i) for each decision z). Knowing the cost function can be a substantially simpler problem than the bandit feedback setting. Originality: Ok The paper extends prior work for learning from logged interventions in two important ways -- what if the decision set is continuous multi-dimensional, and what if the intervention policy is unknown. I expected semi-synthetic experiment comparisons with [16] (which assumed the intervention policy is known), and the straightforward modification of [16] with imputed generalized propensity scores to address the paper's problem setting. Significance: Good Several real-world problems (e.g. the semi-synthetic setup for Warfarin dosing) can benefit from better algorithms for the problem studied in the paper. Reproducibility: Not clear what hyper-parameters ended up getting selected in each of the two experiments. I would be more confident if there was code accompanying the paper (the datasets are public anyway). I have read the other reviews and the author feedback. I maintain my borderline accept rating for the paper. The paper can be much stronger with a more careful analysis of the V(x,z) and B(x,z) terms in the objective. If there was a toy problem where we knew bias in closed form, do we indeed see that penalizing bias helps (or, is the Lipschitz upper-bound on bias very loose and actually achieving a different regularization)? If we have a problem with heterogenous outcomes, is the homoscedastic V estimate helping or hurting? There are heuristics to get variance estimates from regression models (or even conformal regression methods that provide an estimate and confidence interval). Do they work ok or is the method reliant on estimate of V using MSE on the training set? With such an analysis, the paper will be substantially stronger.

Reviewer 2



This paper explores off-policy learning. It proposes an optimization framework that incorporates information on the uncertainty of the predicted outcomes to learn better policies. It is based on the fact that minimizing only the cost function would result in a low bias but high variance estimator. Their idea is to add the estimator’s variance and bias as penalty terms to the objective function to trade-off between learning estimators with small cost and those with high variance/bias. However, it is not clear whether the findings are novel, as Swaminathan and Joachims [21] also consider incorporating the variance of response into the objective function to attain models that exhibit lower variance. While the authors also mention that Swaminathan and Joachims [21] assume that the underlying mechanism of action selection (i.e., propensities) is known, their proposed Counterfactual Risk Minimization framework is independent of how the response variable (u in [21]) is defined (i.e., whether it requires the propensity score) and therefore does not rely on this assumption. The authors needs to better explain how their work differs from that of [21]. Unfortunately, many parts of the paper are quite hard to follow. The authors should clarify the following points: L22: Is the cost function known/given? e.g., Is it c = (y - \hat{y})^2, where y is the true outcome and \hat{y} is the predicted outcome by a regression fit? Or is it determined arbitrarily? It is very important to define the cost clearly, as the rest of the equations are heavily dependent on it … The experiments do not shed light on what the cost function is either. L117: “The first step …” → Are there any next steps? My understanding is that learning the predictive model is intertwined with optimizing for z since there is nothing else to learn other than c (and consequently w due to the way it’s defined in L149.5) in equation (2). L169-170: If we could find a model that predicts the counterfactual outcomes accurately, we could use that to select the best z; the problem is that, due to sample selection bias (i.e., dependence of z on x), we cannot be sure that the regression fit would give accurate estimates of the counterfactuals. Therefore, it is wrong to base the model selection on this. Unless the above mentioned points are clarified, it is hard to judge the quality of this work. Minor points: L40: parameters are “learned”; objective function is “optimized”. L120-123: Unclear nomenclature: what is the “response variable”? What is “prediction of the algorithm”? and what is the “training response”? L128: “reduces bias of the predictions” → predictions of what? outcome or cost? L142.5: Did you mean B(x,z) := \sum_{i} || \hat{\mu(x,z)} - \mu(x,z) ||_2 ? L160-161: Isn’t this the role of the predictive model? Isn’t this already done? Typos: L124: satisfies → to satisfy L211: probably → probable ==== In light of the authors’ discussion in the rebuttal, I am convinced that the contribution of this paper is novel (compared to [21]). Regarding my L169-170 comment -- although the authors indicated that they get good empirical results: I am still not convinced that this way of tuning parameters is valid. That is, tuning hyperparameters based on predicted counterfactuals, especially when the sample selection bias is not accounted for in prediction procedure. I expect this method to fail in case of a high selection bias in data and suspect that this was not the case in the paper’s experiments. In general, it was difficult to understand some main parts of the submission. I believe the paper would benefit from being modified for more clarity and better flow. Given the author’s comments, I am willing to revise my score to 5.

Reviewer 3



This paper is about learning optimal treatment rules with a continuous treatment, in the potential outcomes setup with ignorability. There is a considerable amount of literature on the two-arm version of this problem; however, the present setup (with a continuous treatment choice) has been less explored. Call mu(x, z) the expected cost of deploying decision z for a sample with covariates x. A simple approach would be to to estimate \hat{mu}(x, z), and then choose \hat{z}(x) to optimize the estimated function. A problem with this approach, however, is that if the rewards from some sub-optimal actions are hard to estimate, we might accidentally pick a very bad action. The proposal of the paper is to penalize actions z(x) for which \hat{mu}(x, z) is very variable or biased and, in case of uncertainty, to prefer actions whose rewards can be accurately estimated. The authors provide both theoretical and empirical backing for this method. Conceptually, the paper builds on work from Maurer & Pontil and Swaminathan & Joachims on uncertainty penalization; the setup here is different in that the data-collection policy may not be known and the action space may be continuous. Overall, this method is principled, and may work well in many areas. The paper is also very well written. One advantage in particular is that, in many situations, we might have most data from the “status quo” policy, and this method will correctly preserve the status quo unless there is strong evidence that another policy is better.